# Influence of Surgical Technique on Post-Operative Complications in the Extraction of the Lower Third Molar: A Retrospective Study

**DOI:** 10.3390/dj11100238

**Published:** 2023-10-17

**Authors:** Massimo Albanese, Alessandro Zangani, Federica Manfrin, Dario Bertossi, Rachele De Manzoni, Nicolò Tomizioli, Paolo Faccioni, Alessia Pardo

**Affiliations:** Dentistry and Maxillofacial Surgery Unit, Department of Surgery, Dentistry, Paediatrics and Gynaecology (DIPSCOMI), University of Verona, Piazzale L.A, Scuro 10, 37134 Verona, Italy; massimo.albanese@univr.it (M.A.); alessandro.zangani@univr.it (A.Z.); dario.bertossi@univr.it (D.B.); rachele.demanzonicasarola@univr.it (R.D.M.); paolo.faccioni@univr.it (P.F.); alessia.pardo@univr.it (A.P.)

**Keywords:** impacted third molar, complications, triangular flap, envelope flap

## Abstract

The surgical extraction of the impacted third molar is frequently associated with several complications. The purpose of this study is to assess how two different surgical protocols affect post-operative complications during the extraction of the lower impacted third molars. In order to compare and evaluate two different techniques (triangular flap vs. envelope flap), and the relative post-extraction complications, two groups of 150 patients each underwent to surgical impacted third molar extraction and 60 days of follow-up. The complication rate in the two groups was 14.00% in group A and 17.33% in group B. There was a strong association between smoking (OR: 2.8) and the use of oral contraceptives (OR: 1.75) with complications. The age- and sex-related incidence of complications in hard tissue healing has great variability in the literature; the analysis performed on our data did not show a statistically significant association between them. Even though related to a higher incidence of transient changes in sensitivity, it was found that the envelope flap saw a lower percentage of complications. There is still no clarity on which is the best protocol for the extraction of the lower impacted third molar, and the choice often depends on the surgeon’s experience.

## 1. Introduction

Mandibular third molars are found in 90% of the general population, with 33% of the people having at least one impacted molar [1,2].

The high prevalence of impacted third molars might be attributed to both genetic and environmental factors [3].

The surgical extraction of impacted mandibular third molars is the most common procedure in oral and maxillofacial surgery [4,5,6], and is frequently associated with considerable postoperative adverse effects [7,8].

The indications for extracting lower third molars include pericoronitis, periodontal defects posterior to the second molars [9], caries of the second or third molars, neurogenic and myofascial pains, odontogenic cysts and tumors, and surgical indications noted with general dentistry, oropharyngeal hygiene, prosthodontic and orthodontic considerations [10].

The surgical technique may present variables such as flap design, bone removal coronectomy, or root separation, and every procedure must be performed without damaging the surrounding anatomical structures [11].

Some of the most frequent postoperative complications in impacted third molar extraction are pain, swelling, trismus, prolonged bleeding, dry socket, alveolar osteitis, surgical site infections, abscesses, and sensory alterations of the inferior alveolar nerve (IAN) or lingual nerve (LN) [5,6,8,10,12,13]. The frequency of these postoperative complications for lower impacted third molars varies in the literature between 0% and 30% [5,14,15,16].

Smoking, female sex, the use of oral contraceptives (OCPs), age, the surgeon’s technique, the level of tooth impaction, and an increase in operating time have been suggested as potential risk factors for postoperative morbidity [17,18,19]. 

The results of the meta-analysis confirmed that the use of oral contraceptives (OCP) is a risk factor for alveolar osteitis (AO) after extraction of the third molar. The lack of a significant difference between females and males who did not take OCPs suggests that female sex is not a risk factor for AO [20].

The incidence of postoperative complications and the risks of permanent sequelae increase with age. Therefore, once the decision to extract an impacted mandibular third molar has been made, the surgery should be performed as soon as possible and well before the age of 24 [6]. Several articles in the literature suggest that neither the design of the flap nor its extension affects postoperative symptoms, signs, and complications [7,21]. 

Instead, many other articles in the literature emphasize how it is essential to note that flap design is a crucial factor in the surgery of an impacted third molar, as it affects the visibility and accessibility of the impacted tooth, as well as the subsequent healing of the surgical defect created by the surgery [2,22]. The envelope flap and triangular flap, both described since the 1950s [23], are the most commonly used flap designs for impacted lower third molar surgeries [11,21,24,25]. Several studies suggest that an envelope flap may reduce postoperative pain and swelling compared to a triangular flap [23,26]; indeed, the conventional envelope flap remains the gold standard for the exposure and removal of impacted lower third molars [27]. The triangular flap was more efficacious when we considered the postoperative periodontal status of the adjacent second molar and the dehiscence following wound healing in comparison to those with the envelope flap [22]. 

During tooth extraction surgery, the manipulation of hard and soft tissues for mucoperiosteal flap reflection and subsequent bone removal involves various factors that can affect the patient’s postoperative course in terms of pain, swelling, trismus, and healing. In this context, the choice of surgical access flap can affect the postoperative outcomes of third molar surgery, including the development of numerous complications [13,22]. 

The aim of this study is to examine how two different surgical protocols impact post-operative complications during the extraction of lower impacted third molars.

Were assessed whether or not the type of flap, comparing the envelope flap and the triangular flap, does affect the incidence of complications.

## 2. Materials and Methods

In this study, patients underwent surgical extractions of the lower impacted third molar from March 2013 to March 2019, at the Unit of Maxillo-Facial Surgery and Dentistry of the University Hospital of Verona (Italy), and the surgeries were performed by two operators, M.A. and D.B. 

Based on the medical history of the patients who were eligible for surgery, the exclusion criteria were as follows: systemic diseases and/or drug therapies known to interfere with soft and hard tissue healing, smokers, pregnant women, and patients with chronic periodontal disease. In addition, those who had undergone oral and/or periodontal surgery during the previous 12 months were excluded. Only patients with a healthy dental and medical status were included in the study. 

Based on the inclusion criteria, 300 patients aged between 11 and 80 years were enrolled in the study; all patients complied up until the follow-up period of 60 days. The demographic data were recorded, thorough history was taken, and informed written consent was obtained from the patients.

Patients were examined with cone beam computed tomography examination before surgery, to identify the relationship of the tooth to the inferior alveolar nerve [28]. Moreover, the third molars were classified in accordance with Pell and Gregory (1933) [29], and those classified as Class IA were excluded from the study; radiographical inclination and root form were not considered exclusion criteria. 

Patients undergoing surgery were divided into two groups, according to which the operator had performed the extraction of the impacted third molars. Each operator performed the extraction with a different type of flap: operator A with the Triangular Flap, operator B with the Envelope Flap.

The nature and aim of the study together with anonymity in the scientific use of the data were clearly presented in a written informed consent form, obtained from all subjects involved in the study; appropriate forms were also regularly filled in by parents or related legal guardians for patients < 18 years. The study was also conducted in accordance with the Declaration of Helsinki and good clinical practice guidelines for research on human beings. The University Institutional Review Board approved the retrospective study (Protocol “POST-ESTR.LEMBI”, Prog. 3921CESC, 3 August 2022). The study is in compliance with the STROBE checklist guidelines.

### 2.1. Pre-Surgical Procedures

Patients were given antibiotic prophylaxis via two tablets of 875 mg amoxicillin + 125 mg clavulanic acid 1 h before surgery. Before surgery, patients were rinsed with 0.20% Chlorhexidine for 1 min.

### 2.2. Surgical Procedures

An amount of 3.6 mL of mepivacaine HCl 2% with 1:100,000 epinephrine was used as the local anesthetic agent for inferior alveolar and lingual nerve block (1.8 mL), in association with a buccal injection (1.8 mL). In the case of blood aspiration during the injection for inferior alveolar nerve block, the vial was changed and the inoculation site of the anesthetic was slightly varied.

In patients undergoing protocol A, the procedure was performed with a triangular flap. An incision was made from the anterior border of the mandibular ramus to the distal surface of the distobuccal cusp of the mandibular second molar. It was extended along the buccal sulcus up to the medial corner of the second molar crown. The relieving medial vertical incision was oblique to the mandibular buccal fornix, and aligned with the mesiobuccal cusp of the second molar. A full thickness flap was then elevated. Osteotomy was performed with a rose head bur and odontotomies were performed with a fissure bur by using a surgical straight handpiece. Finally, the tooth was extracted with the use of levers of Bein and forceps.

In patients undergoing protocol B, the procedure was performed with an envelope flap. An incision was made from the anterior border of the mandibular ramus to the distal surface of the distobuccal cusp of the mandibular second molar. It was extended along the sulcus to the distobuccal corner of the second and first molar crown. The incision was continuous, without being relieved. A full-thickness flap was then elevated. Osteotomy was performed with a rose head bur to expose the dental crown, and then the tooth was removed with levers of Bein and forceps without odontotomy. 

After tooth extraction, bony irregularities were corrected, the post-extraction site was curetted with a Volkmann spoon and rinsed with 0.9% (125 mL) saline solution. Surgical wounds were closed using braided absorbable sutures (Vicryl 4.0; Ethicon, Somerville, NJ, USA).

### 2.3. Post-Surgical Procedures

Antibiotic therapy was prescribed with 875 mg amoxicillin + 125 mg Clavulanic acid tablets every 8 h for 5 days.

Sutures were removed 7 days after surgery. 

During the post-operative observation period, we detected delayed healing, alveolitis, osteitis, and osteomyelitis according to the timing of onset and the duration of complication.

### 2.4. Statistical Analysis

The two cohorts were merged into a single database and were analyzed using Stata v.15.1 (Statacorp., College Station, TX, USA). The variables considered were all categorical (or binary) except that age was divided into two classes (age < 25 and age ≥ 25). The chi-square test compared the percentages obtained from both cohorts on the variables sex, age, smoking, and oral contraceptive intake to verify that there was no statistically significant difference between the two cohorts, so that the two homogeneous patient pools could be defined. Complication rates in the two cohorts were then compared using the Fisher test. Through logistic linear regressions, the strength of association between the variables sex, age, smoking, and contraceptives with the percentages of complications was analyzed. The significance level was set at 0.05.

## 3. Results

Procedural times were analyzed for both groups, reporting an average surgical time of more than 45 min for Protocol A and that of less than 45 min for Protocol B; the average osteotomy time was 1′26″ for Protocol A and 0′36″ for Protocol B.

The two groups comprised 150 patients each; 54.67% of patients operated on under protocol A were female and 45.33% of them were male, while 48.00% of those operated on under protocol B were female and 52.00% of them male. The average age of patients was 28 for group A and 26 for group B. The complication rate for the two groups was 14.00% in group A and 17.33% in group B. Since no osteomyelitis was found in either patient pool, the proportion of osteomyelitis (% = 0) was omitted from the tables below (Table 1).

The percentages of complications were calculated by dividing the two populations into two subgroups based on an age of 25 years or older (Table 2).

### 3.1. Group A

Smoking patients comprised 23.33% (35 patients) of the group, of whom 17 were female and 18 were male. In the patient pool, there were 13 females (8.67%) taking oral contraceptives and 3 of these were also smokers (2.00%). The percentages of complications in these sub-groups were calculated and it was found that in female smokers the % of complications was 29.41% (5/17 patients); in non-smoking females the % of complications was 12.30% (8/65 patients); in females taking oral contraceptives the % of complications was 23.08% (3/13 patients); in females taking oral contraceptives and smoking the % of complications was 33.33% (1/3 patients); in smoking males the % of complications was 22.22% (4/18 patients); in non-smoking males the % of complications was 8.00% (4/50 patients).

### 3.2. Group B

In total, 26.66% (40 patients) in the group were smokers, 16 of whom were female and 23 were male. Briefly, 16 females (10.67%) patients were taking oral contraceptives and of these 7 were also smokers (4.66%). The percentage of complications in the female smokers group was 50.00% (8/16 patients); in non-smoking females the % of complications was 10.71% (6/56 patients); in females taking oral contraceptives the % of complications was 31.25% (5/16 patients); in females taking oral contraceptives and smoking the % of complications was 42.86% (3/7 patients); in male smokers the % of complications was 26.09% (6/23 patients); in non-smoking males the % of complications was 10.90% (6/55 patients).

### 3.3. Statistical Analysis

The average age of group A patients was 27.9 ± 14.07 with the youngest being 12 years old and the oldest being 80 years old; the average age of group B was 26.2 ± 12.00 with the youngest being 11 years old and the oldest being 74 years old.

The chi-square test showed that the two cohorts of patients were evenly distributed in terms of sex, smoking habit, age, and oral contraceptive intake (Table 3).

Using the Fisher test, the total percentages of dry alveolites, purulent alveolitis, and osteitis in the two groups were compared and no statistical significance was found (*p*-value = 0.732). 

Taking into account, at moment of extraction, non-smoking patients and those that did not take oral contraceptives, by comparing the percentages of dry and purulent alveolites and osteitis, there was no statistically significant difference between the two groups (*p*-value = 0.339).

Linear regressions showed the strength of association between smoking and oral contraceptives with complications. The strength of association between risk factors and complications was calculated. The odds ratio for smoking was 2.8, while that for oral contraceptives was 1.75; analyzing the association of the two risk factors, the odds ratio was 3.53. Gender (*p*-value = 0.599) and age (*p*-value = 0.952) were not found to be significantly associated with complications in this study.

Using the Fisher test, we compared the percentages of paresthesia in the two cohorts (0% in group A and 5.33% in group B), and the difference was statistically significant (*p*-value = 0.007).

## 4. Discussion

The purpose of this study is to evaluate the influence of the surgical protocol used on the healing complications after lower impacted third molar extraction comparing the envelope flap and the triangular flap. Based on the time of onset and resolution, the rates of delayed healing, alveolitis sicca dolorosa, dry alveolitis, purulent alveolitis, osteitis, and osteomyelitis were evaluated in this study. However, these parameters had to be assigned arbitrarily because the literature does not offer specific documentation on delayed healing and does not separate osteitis from alveolitis; as regards osteomyelitis, only case reports were found as available documentation, given the rarity of its frequency. In this study, no osteomyelitis was found in either pool.

The patients were checked following the extraction of the lower impacted third molars at 1, 3, 5, 7, 14, 28 days, and 60 days; the sutures were removed 7 days after surgery. During post-operative examinations, the healing condition of the soft tissues, the presence of pain, suppuration, and the absence or presence of sensory changes in the territories innervated via IAN and LN homolateral extraction were detected. The main limitations of this study are the small sample, the 60-day follow-up, the performance of extractions by experienced surgeons with the techniques used in a university hospital environment, and the inclusion of only healthy patients.

It is important to note that the impacted third molars of this study were extracted between March 2013 and March 2019. Prophylactic antibiotics were provided to all patients, even if they were healthy. In the latest guidelines, antibiotic prophylaxis is recommended only for patients with a high risk of severe infectious endocarditis [30,31,32].

During post-operative examinations, according to the state of alterations in healing, diagnoses were made with the following criteria:-Delayed healing: not well-bound red soft tissue was still present when the suture was removed; the situation was normalized at the next control.-Alveolitis: from the first to the fifth day after surgery, there were signs of infection of the post-extractive site, with or without associated pus; the resolution occurred with appropriate treatment within 20 days.-Osteitis: episodes of pain and infection of the site after 30/40 days, associated with restitutio ad integrum within two months of surgery.-Osteomyelitis: episodes of pain and infection for more than two months after the extraction operation, associated with the pathological sign of bone seizure.-Paresthesia: the appearance of local sensitivity alterations, associated with numbness, tingling, or burning. This may be temporary or permanent.

The present study had some limitations, including the sample size. In addition, during the analysis of complications the parameters of swelling, edema, reduction in mouth opening, inflammation in the masticatory muscles, hematoma, and pain were not examined, firstly because they were considered physiological in the post-operative course and also because these signs and symptoms are far more complicated due to the number of measures needed and the tendency of errors to occur.

### 4.1. Alveolitis

Post-extraction alveolitis (sicca or purulent) is one of the most frequent complications after a surgical extraction.

It can arise at the site of any dental element, but is much more frequent at the level of the lower third molar, and it is often referred to as “post-extraction complication of the lower third molar” [33,34,35]. The incidence of this after an extraction of the lower molar is 1% to 68.4% [6,34,36].

In this study, which analyzed only lower impacted third molars, the prevalence of alveolitis was found to be 6.00% in group A patients and 5.33% in group B patients.

### 4.2. Osteitis

The incidence of alveolar osteitis in the literature ranges from 0.4% to 17% [10].

In our study, the incidence of osteitis was 2.67% in group A and 0.67% in group B. The most affected population was that of females, and as regards age, those under 25 seemed to be most affected.

When considering the initial phases of healing, AO can be considered a relatively frequent complication; in particular, the incidence of AO was greater in the lower third molar when compared with that in maxillary [13,37].

### 4.3. Age

Patients were divided according to age into two groups (<25 and ≥25 years). 

The pool of the first operator saw an incidence of 1.33% of healing delays and 2.00% of alveolitis in patients under the age of 25; an incidence of 4.00% of delayed healing and 4.00% of alveolitis in patients was observed in patients aged 25 years or older. 

In the pool of the second operator, the incidence of healing delays was found to be 4.00% and that of alveolitis was 3.33% in patients under the age of 25; on the other hand, in patients aged 25 years or older, the incidence of healing delays was 2.00%, while that of alveolitis was 2.00%. 

In support of the above, the age-related incidence of complications related to hard tissue healing is shown to have great variability in the various studies in the literature [33,38,39], and the statistical analysis performed on the data of these two cohorts under examination did not show a statistically significant association strength (*p* = 0.952).

We can consider age a risk factor from the physiological point of view due to increased bone density, complete root formation, and reduced healing capacity, but we must not underestimate the power of association with defined confounding factors, such as a cigarette smoking habit, the use of oral contraceptives in women, poor oral hygiene due to decreased interest compared to oral hygiene the younger population.

### 4.4. Sex

When patients were divided into males and females, the complication rate in group A was 8.67% in females and 5.33% in males; in group B, this was 9.33% in females and 8.00% in males.

These values are in agreement with those in the literature as several authors report that it is estimated that alveolitis occurs twice as often in the females as they do in males [6,40,41].

Female sex is reported in the literature as a risk factor for AO. The correlation is still controversial and the use of OCP remains a confounding factor of considerable impact [20].

Despite the evidence reported in the literature, in our study, probably due to the limited number of patients, the statistical analysis carried out through logistic regression did not show a significant association between sex and complications (*p* = 0.599).

### 4.5. Smoking

Therefore, we wanted to investigate some of the most well-known confounding factors, the first being cigarette smoking. In the pool of the first operator, the percentage of smokers was 26.47% for males and 20.73% for females; for group B, the percentage of smokers was 29.49 for males and 22.22% for females. In both groups, the incidence of complications significantly increased when a smoking habit was present.

In agreement with the literature, we can also state that cigarette smoking is one of the factors that significantly influences the healing process after surgery [42,43], and, to confirm this, the statistical regression test demonstrated that there is a significant association strength between smoking and hard tissue complications (*p* = 0.004). 

### 4.6. Oral Contraceptives

Another risk and confounding factor evaluated is the use of the contraceptive pill in the group of female patients of both operators. Many authors agree that female hormones, in particular estrogen, affect healing, which is why in addition to the use of the pill, the period of the menstrual cycle during which the extraction is performed also seems to lead to different results [44].

In the study by Almeida et al., the difference in the incidence of post-operative complications between women taking contraceptives and women not taking contraceptives was statistically significant (37.9% vs. 8.9%) [45].

Eshghpour et al. revealed statistical significance from their data with an incidence of 24.2% in women who took contraceptives compared to that of 11.5% in women who did not [46].

In accordance with the literature, in this study the incidence was found to be 23.08% in the pool of the first operator and 31.25% in the pool of the second.

The exact mechanism by which estrogens influence the healing process is not yet fully understood, but as early as in the 1960s it was discovered that they were involved in the mechanism of fibrinolysis; in particular, it was seen that they were able to indirectly activate the fibrinolytic system (by increasing the production of factor II, VII, VIII, and X, and plasminogen) and consequently increase blood clot lysis [47].

A correlation between female hormones and the incidence of post-extraction complications has therefore been observed, and several studies have verified the association [48]. It can probably be said that risk factors such as sex and age must be questioned as there are different variables that can act as confounding factors and some of these have high-enough scientific relevance to be bypassed.

### 4.7. Association of Smoke and Oral Contraceptives

The analysis of these factors was concluded by verifying how much the incidence increased by combining the smoking risk factor and the contraceptive risk factor; the reported incidences are 33.33% and 42.86%, according to the latest literature reviews such as that of Taberner et al. and that of Mamoun et al. [16,49].

From the statistical analysis, the strength of association of these two combined factors was found to be statistically significant (*p* = 0.016) with an odds ratio of 3.53. This means that a woman who takes an oral contraceptive and smokes has more than three times the risk of undergoing a pathological recovery process. 

### 4.8. Flap Design

According to two recent systematic reviews, by Da Silva et al. and De Marco et al., there are no statistically significant differences regarding postoperative clinical morbidities when comparing the use of different access flaps for impacted third mandibular molar surgery. Cumulative evidence suggests that flap selection is related to surgical difficulty, which is mainly determined via the location of the tooth to be extracted. Therefore, the dental surgeon chooses the technique based on their clinical experience or surgical preference [13,21].

The ideal flap should provide good visibility and accessibility to the tooth, with minimal impact on adjacent structures.

Important variables in the postoperative course after a surgery of the impacted third molar are the degree and type of inclusion, the amount of bone to be removed, and the time spent for the procedure, which is the most influential [27].

### 4.9. Overheating of Bone 

Defining the two surgical protocols, it can be seen that operator A uses a round bur mounted on a straight handpiece at a speed of 400,000 rpm in a relevant way, freeing the dental element from the encumbrance of the surrounding bone as much as possible in order for it to be easily extracted at a later time with the use of pliers. Often, the operator resorts to odontotomy, which is the splitting of the crown first and then that of the roots of the tooth, with a diamond bur also mounted on a straight handpiece, and then proceeds with removal with minimal effort in using the lever and forceps. Operator B, on the other hand, also uses a round bur mounted on a straight handpiece at a speed of 400,000 rpm for the osteotomy, but in a much more “conservative” way; that is, the operator removes the minimum amount of bone that will allow them to insert a lever and make it strong. The use of odontotomy is reserved for rather rare cases, when, for example, the roots surround the inferior alveolar nerve and then merge below it (in this study, out of 150 cases, only one odontotomy was applied). The force transmitted via the lever much higher than that with the previous technique for obvious reasons; this operative method is reported in the literature under the name of “buccal approach techniques” and allows a minimization of the osteotomy, which is the opposite case to that of the ‘rotary instrument technique’.

The literature agrees that one of the major causes that lead to the pathological healing of hard tissues after a dental extraction is the use of rotary instruments or, to be precise, the heat generated by them. In particular, it has been seen that, during the removal of bone tissue, the resistance produced by the cortex causes an increase in the temperature of the bone itself, via the generation of what is called frictional heat [50,51].

What results seems to be precisely “thermal bone necrosis” [52]. 

Several studies have investigated what the threshold is and most agree that a temperature above 47 °C that lasts for more than 1 min leads to irreversible cell damage and a permanent replacement of bone by fat [53]. 

To decrease the temperature is fundamental irrigation, usually performed with saline solution, thanks to which it is possible for the temperature to stay below the limit temperature. However, in an animal study by Morris et al. in 1985, it was seen that the temperature of the tissues and the duration of exposure to that temperature are linked via an “Arrhenius relationship”, a linear differential equation. In fact, this study showed that for temperatures above 42.5 °C, for every 1 °C increase in temperature, half of the exposure time (factor 2) was enough to have the same biological effects [54].

Given that the use of simultaneous irrigation is now an undisputed protocol for lowering temperature, it is considered necessary in order to minimize the possibility of creating damage to focus on the duration of the surgery, in particular that of the osteotomy. As early as 1975, Horton et al. concluded in his study that healing in the post-extraction sites of dogs appeared to be histologically better when avulsion was performed with the use of levers than when it was performed with the use of burs [55].

Thus, in the study by Singh et al. in 2013, which compared the avulsion technique with levers to that with rotating water instruments, and it was shown, in agreement with this study, that the incidence of alveolitis was higher in the second group [56].

Further supporting evidence found in terms of this is the different average surgical timings of an osteotomy, which are 1′26″ for operator 1 and 0′36″ for operator 2.

### 4.10. Paresthesia

The incidence of lesions of the lower alveolar nerve varies in the literature between 0.4% and 8.4% [6,10,57].

The literature’s index for lingual nerve lesions is 0.2% to 2%, but several studies show that lesions are almost entirely temporary damage with total symptom resolution within 12–13 weeks [10,12,58].

Permanent damage seems to be very rare.

Furthermore, it is appropriate to emphasize the difference in the incidence of paresthesia in the two groups of patients.

The collected data revealed an incidence of 5.34% of paresthesias in pool 2, compared to an incidence of 0% in pool 1.

Briefly, 5.34% is equivalent to eight patients and in particular four had paresthesias of the NAI (inferior alveolar nerve) while four cases pertained to the LN (lingual nerve). These data were also subjected to statistical analysis, using the chi-square test, and there was statistical significance. All reported paresthesias were temporary, and at 60 days, no patients showed symptoms of transient nerve damage. 

The triangular flap allows a better surgical view of the tooth and less traumatic extraction; this may lead to a lower number of paresthesias.

In support of this, in the study by Singh et al., there was also a higher incidence of paresthesia using the lever shown, compared to that when osteotomy with burs was performed [56].

The rationale behind this cause–effect correlation, the use of levers for paresthesia, is not fully clarified in the literature. Nowadays, it is known that one of the factors that can lead to temporary paresthesia is pressure exerted via a blood clot in those third molars with roots that are very close to the mandibular canal. Perhaps, as pressure is a confirmed causal factor in complications related to nerve structures, the vigorous and marked/ long-lasting use of the lever transmits pressure to the nerve that leads to stretching and consequently to paresthesia, in the same way that the blood clot acts.

Given the limitations of the studies reported in the literature, further evaluations are necessary in order to affirm with more certainty and scientific support the correlation between the use of a lever and damage, albeit temporary, to nervous structures.

In addition, some clinicians in the literature have shown that there are fewer complications in very complex extractions of the lower third molar associated with the position of the alveolar nerve very close to the root, warranting a coronectomy, while intentionally leaving the roots in the cavity [59]. Cosola et al. reported that in 130 patients who underwent this minimally invasive surgical procedure with an average follow-up of 4 years, only 13 roots had the complications of moving coronally relative to the starting position, but all were asymptomatic and, in some cases, underwent a second extraction but without a risk of displacement with respect to the nerve [60].

## 5. Conclusions

The objective of this study was to assess whether or not there is a difference, regarding post-operative complications, after impacted third molar extraction, comparing two surgical protocols: one using a triangular flap and the other using an envelope flap. Comparing the two groups, the only statistically significant difference (*p*-value = 0.007) was alterations in sensitivity, with an incidence of 0% for the triangular flap and 5.33% for the envelope flap. Although related to the increased incidence of transient changes in sensitivity, the envelope flap led to 12% of complications occurring in hard and soft tissue healing, which is lower than the 14% of complications observed with the triangular flap. In conclusion, when comparing the two types of flap, in with the envelope flap there were more paresthesias and less complications, while with the triangular flap there was no paresthesia but more complications in the healing of hard and soft tissues. In the literature, there is still no consensus on which the best protocol for the extraction of the lower impacted third molars is; nevertheless, this study, with its limitations, seems to show a lower frequency of complications when performing surgery with the envelope flap.

## Figures and Tables

**Table 1 dentistry-11-00238-t001:** Complications sorted by sex.

Complications by Sex Group A	Female		Male		Total	
	%	n	%	n	%	n
Purulent alveolitis	0.00%	0	1.33%	2	1.33%	2
Alveolitis sicca dolorosa	4.00%	6	0.67%	1	4.67%	7
Alveolar osteitis	2.00%	3	0.67%	1	2.67%	4
Delayed healing	2.67%	4	2.67%	4	5.33%	8
Total	8.67%	13	5.33%	8	14.00%	21
**Complications by Sex** **Group B**						
	%	n	%	n	%	n
Purulent alveolitis	1.33%	2	2.00%	3	3.33%	5
Alveolitis sicca dolorosa	0.00%	0	2.00%	3	2.00%	3
Alveolar osteitis	0.67%	1	0.00%	0	0.67%	1
Delayed healing	3.33%	5	2.67%	4	6.00%	9
LN paresthesia *	2.00%	3	0.67%	1	2.67%	4
IAN paresthesia *	2.00%	3	0.67%	1	2.67%	4
Total	9.33%	14	8.00%	12	17.33%	26

* IAN: inferior alveolar nerve; LN: lingual nerve.

**Table 2 dentistry-11-00238-t002:** Complications sorted by age.

Complications by Age Group A	Over 25		Under 25		Total	
	%	n	%	n	%	n
Purulent alveolitis	0.67%	1	0.67%	1	1.33%	2
Alveolitis sicca dolorosa	3.33%	5	1.33%	2	4.67%	7
Alveolar osteitis	0.00%	0	2.67%	4	2.67%	4
Delayed healing	4.00%	6	1.33%	2	5.33%	8
Total	8.00%	12	6.00%	9	14.00%	21
**Complications by Age** **Group B**						
	%	n	%	n	%	n
Purulent alveolitis	1.33%	2	2.00%	3	3.33%	5
Alveolitis sicca dolorosa	0.67%	1	1.33%	2	2.00%	3
Alveolar osteitis	0.00%	0	0.67%	1	0.67%	1
Delayed healing	2.00%	3	4.00%	6	6.00%	9
LN paresthesia	1.33%	2	1.33%	2	2.67%	4
IAN paresthesia	2.00%	3	0.67%	1	2.67%	4
Total	7.33%	11	10.00%	15	17.33%	26

**Table 3 dentistry-11-00238-t003:** Chi-square test.

		Group A		Group B		Statistical Significance
		n	%	n	%	*p*-Value
Sex	Male	68	45.33%	78	52.00%	0.202
	Female	82	54.67%	72	48.00%	
Age	Under 25	82	54.67%	88	58.67%	0.413
	Over 25	68	45.33%	62	41.33%	
Smoke	Yes	35	23.33%	40	26.66%	0.592
	No	115	76.67%	110	73.34%	
ACP	Yes	13	15.85%	16	22.22%	0.293
	No	69	84.15%	56	77.78%	

## Data Availability

Data are available upon reasonable request to the corresponding authors.

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
