# Peer review of "Influence of Surgical Technique on Post-Operative Complications in the Extraction of the Lower Third Molar: A Retrospective Study"

_dentistry, 2023, doi:10.3390/dj11100238_

Round 1

Reviewer 1 Report

Dear colleagues!

Your research is done on a good methodological level. But there are key issues that are important to highlight.

1. You need to specify the null hypothesis of the study

2. In the Materials and Methods section, you need to enter information about the calculation of the sample size, as well as information about the number of patients, inclusion and exclusion criteria from the study.

3. Why were all patients prescribed the same antibiotic therapy? Did they all have the same status or do you not practice an individual approach?

You have the same antibiotic both preoperatively and postoperatively. In fact, there is no break and there is no need to single out separate stages.

4. Lines 102-110 are highlighted and underlined. Check the correctness of this paragraph.

5. How was anesthesia performed, by what method and what were the results of the aspiration test?

6. In the results you write about paresthesia - what were the prerequisites for this complication, what prevention methods were made? Have you analyzed CT scans of patients?

7. You need to update the list of references, add more recent studies of the last 5 years and remove / reduce the number of those that are more than 20 years old

Author Response

REVIEWER 1

Dear reviewer,

Thank you for your contribution.

Below are the answers to your suggestions.

  1. You need to specify the null hypothesis of the study

Line 78. The null hypothesis has been specified: in order to assess whether the type of flap does not affect the incidence of post-operative complications.

  1. In the Materials and Methods section, you need to enter information about the calculation of the sample size, as well as information about the number of patients, inclusion and exclusion criteria from the study.

Lines 86-91: Based on the medical history of the patients who were eligible for surgery, exclusion criteria were: systemic diseases and/or drug therapies known to interfere with soft and hard tissue healing, pregnant women, patients with chronic periodontal disease. In addition, those who undergone oral and/or periodontal surgery during the previous 12 months were excluded. Only patients with healthy dental and medical status were included in the study.

  1. Why were all patients prescribed the same antibiotic therapy? Did they all have the same status or do you not practice an individual approach?

You have the same antibiotic both preoperatively and postoperatively. In fact, there is no break and there is no need to single out separate stages.

All patients in the study were, as described in the inclusion criteria /exclusion healthy patients; antibiotic therapy was the same for all to not create variables that could affect the study.

  1. Lines 102-110 are highlighted and underlined. Check the correctness of this paragraph.

The error was a residue of revision among the authors, it was fixed formatting.

  1. How was anesthesia performed, by what method and what were the results of the aspiration test?

Lines 118-120: 6 ml of mepivacaine HCl 2% with 1:100,000 epinephrine used as the local anesthetic agent for inferior alveolar and lingual nerve block (1,8 ml), In association with a buccal injection (1,8 ml).

Added: In case of blood aspiration during the injection for inferior alveolar nerve block, the vial was changed and the inoculation site of the anesthetic was slightly varied.

  1. In the results you write about paresthesia - what were the prerequisites for this complication, what prevention methods were made? Have you analyzed CT scans of patients?

Lines 94: Patients were examinated with cone beam computed tomography examination before surgery[28].

Added: to identify the relationship of the tooth with the inferior alveolar nerve.

  1. You need to update the list of references, add more recent studies of the last 5 years and remove / reduce the number of those that are more than 20 years old

The bibliography has been updated. The oldest articles are, however, articles of high scientific value and necessary to discussion.

Reviewer 2 Report

Dear authors the article is very interesting so I am honored to review it. Please add some literature and minor revision in the article.

Introduction:

Please improve the introduction and complication of the surgery of third molar extractions, you have to talk about the antibiotics needs for complex extractions. Moreover, some clinicians in litterature showed that there are less complications in very complex inferior third molar extractions when the position of the alveolar nerve is very near to the root of third molar, doing a coronectomy, I mean intentionally leaving the roots in the socket. This particular surgical procedure is called coronectomy of third molars, please mention about it for example citing: Cosola et al. They reported that in 130 patients who underwent this mini-invasive surgical procedure with 4 years of mean follow-up, just 13 roots had the complications of moving coronally the remanent root from the starting position, but all were asymptomatic and in some cases they receive a second extraction but with no risk because the roots are not anymore near the nerve.
Please add these sentences citing the following 2 articles:

Cosola S, Kim YS, Park YM, Giammarinaro E, Covani U. Coronectomy of Mandibular Third Molar: Four Years of Follow-Up of 130 Cases. Medicina (Kaunas). 2020 Nov 27;56(12):654. doi: 10.3390/medicina56120654.

Cervera-Espert J, Pérez-Martínez S, Cervera-Ballester J, Peñarrocha-Oltra D, Peñarrocha-Diago M. Coronectomy of impacted mandibular third molars: A meta-analysis and systematic review of the literature. Med Oral Patol Oral Cir Bucal. 2016 Jul 1;21(4):e505-13. doi: 10.4317/medoral.21074.

Materials

In the first sentence add the date of ethical committee and specify the number of surgeons.

Discussion and conclusion

These strong disagreements with other studies and this interesting conclusion must be softened by adding more detailed information about the limitations of the present study in the discussion.

Author Response

REVIEWER 2

Dear reviewer,

Thank you for your contribution.

Below are the answers to your suggestions.

Introduction:

Please improve the introduction and complication of the surgery of third molar extractions, you have to talk about the antibiotics needs for complex extractions. Moreover, some clinicians in litterature showed that there are less complications in very complex inferior third molar extractions when the position of the alveolar nerve is very near to the root of third molar, doing a coronectomy, I mean intentionally leaving the roots in the socket. This particular surgical procedure is called coronectomy of third molars, please mention about it for example citing: Cosola et al. They reported that in 130 patients who underwent this mini-invasive surgical procedure with 4 years of mean follow-up, just 13 roots had the complications of moving coronally the remanent root from the starting position, but all were asymptomatic and in some cases they receive a second extraction but with no risk because the roots are not anymore near the nerve.

Please add these sentences citing the following 2 articles:

Cosola S, Kim YS, Park YM, Giammarinaro E, Covani U. Coronectomy of Mandibular Third Molar: Four Years of Follow-Up of 130 Cases. Medicina (Kaunas). 2020 Nov 27;56(12):654. doi: 10.3390/medicina56120654.

Cervera-Espert J, Pérez-Martínez S, Cervera-Ballester J, Peñarrocha-Oltra D, Peñarrocha-Diago M. Coronectomy of impacted mandibular third molars: A meta-analysis and systematic review of the literature. Med Oral Patol Oral Cir Bucal. 2016 Jul 1;21(4):e505-13. doi: 10.4317/medoral.21074.

Added in discussion, lines 460-467: In addition, some clinicians in the literature have shown that there are fewer complications in very complex extractions of the lower third molar associated with the position of the alveolar nerve very close to the root, making a coronectomy, intentionally leaving the roots in the cavity. Cosola et al. reported that in 130 patients who underwent this minimally invasive surgical procedure with 4 years of average follow-up, only 13 roots had the complications of moving coronally relative to from the starting position, but all were asymptomatic and, in some cases, receive a second extraction but without risk for the displacement with respect to the nerve.

Materials

In the first sentence add the date of ethical committee and specify the number of surgeons.

Lines 84-85:and the surgeries were performed by two operators, M.A. and D.B.

Added date of the ethics committee: 03/08/2022

Discussion and conclusion

These strong disagreements with other studies and this interesting conclusion must be softened by adding more detailed information about the limitations of the present study in the discussion.

It was fixed

Reviewer 3 Report

  • Dear Authors,

I have completed my evaluation of the article. 

Here are my comments:

  1. Please note that the study was performed by two investigators. And delete by two different …..

  2. Please rewrite this paragraph: Patients with systemic diseases and/or drug therapies known to interfere with soft and hard tissue healing following surgery, have been excluded from the study. The exclusion criteria were pregnancy, have not undergone oral and/or periodontal surgery during the previous 12 months, patients with chronic periodontal disease. Were included non smokers and patients with healthy dental status. 

  3. In the material and method part, it was mentioned that the sutures were removed 14 days after surgery, whereas in the discussion, they were removed 7 days after surgery. Please clarify this issue.

  4. The tables are informative and well written. 

  5. Please add references to this paragraph: A correlation between female hormones and the incidence of post-extraction complications has therefore begun, and several studies have verified the association. It can probably be said that risk factors such as sex and age must be questioned as there are different variables that can act as confounding factors and some of these have a high scientific relevance to be able to bypass them.

  6. Please delete the years after the authors and rewrite this paragraph: According to two recent systematic reviews, Da Silva et al. (2020) and De Marco et al. (2021), there are no statistically significant differences regarding postoperative clinical morbidities when comparing the use of different access flaps for third mandibular molar surgery. Cumulative evidence suggests that flap selection is related to surgical difficulty which is mainly determined by the location of the tooth to be extracted. Therefore the dental surgeon chooses the technique based on his clinical experience or surgical preference [13,21].

  7. Please rewrite this part: In support of this, also in the study by Singh et al. a higher incidence of paraesthesia using the lever was shown, compared to when the osteotomy with burs was performed [52].

  8. The conclusion should be rewritten.

Minor editing of English language required.

Author Response

REVIEWER 3

Dear reviewer,

Thank you for your contribution.

Below are the answers to your suggestions.

Please note that the study was performed by two investigators. And delete by two different …..

It was verified and corrected

Please rewrite this paragraph: Patients with systemic diseases and/or drug therapies known to interfere with soft and hard tissue healing following surgery, have been excluded from the study. The exclusion criteria were pregnancy, have not undergone oral and/or periodontal surgery during the previous 12 months, patients with chronic periodontal disease. Were included non smokers and patients with healthy dental status.

Rewritten: Based on the medical history of the patients who were eligible for surgery, exclusion criteria were: systemic diseases and/or drug therapies known to interfere with soft and hard tissue healing, pregnant women, patients with chronic periodontal disease. In addition, those who undergone oral and/or periodontal surgery during the previous 12 months were excluded. Only patients with healthy dental and medical status were included in the study.

In the material and method part, it was mentioned that the sutures were removed 14 days after surgery, whereas in the discussion, they were removed 7 days after surgery. Please clarify this issue.

Sutures were removed at 14 days, error within text was corrected

The tables are informative and well written.

Please add references to this paragraph: A correlation between female hormones and the incidence of post-extraction complications has therefore begun, and several studies have verified the association. It can probably be said that risk factors such as sex and age must be questioned as there are different variables that can act as confounding factors and some of these have a high scientific relevance to be able to bypass them.

It was fixed

Please delete the years after the authors and rewrite this paragraph: According to two recent systematic reviews, Da Silva et al. (2020) and De Marco et al. (2021), there are no statistically significant differences regarding postoperative clinical morbidities when comparing the use of different access flaps for third mandibular molar surgery. Cumulative evidence suggests that flap selection is related to surgical difficulty which is mainly determined by the location of the tooth to be extracted. Therefore the dental surgeon Chooses the technique based on his clinical experience or surgical preference [13,21].

It was fixed

Please rewrite this part: In support of this, also in the study by Singh et al. a higher incidence of paraesthesia using the lever was shown, compared to when the osteotomy with burs was performed [52].

Rewritten: In the study by Singh et al., during the extraction of third molars, a higher incidence of paraesthesia was shown when using the lever, compared to when the osteotomy was performed with burs [52].

The conclusion should be rewritten.

The conclusions have been rewritten.

Reviewer 4 Report

Dear Authors,

I have read your manuscript entitled: "Influence of surgical technique on post-operative complications in the extraction of the lower third molar: A retrospective study."

"Patients with systemic diseases and/or drug therapies known to interfere with soft and hard tissue healing following surgery, have been excluded from the study. The exclusion criteria were pregnancy, have not undergone oral and/or periodontal surgery during the previous 12 months, patients with chronic periodontal disease. Were included non-smokers and patients with healthy dental status. [...] Patients were given antibiotic prophylaxis with 875 mg amoxicillin + 125 mg clavulanic acid two tablets 1 hour before surgery."

According to 2020 ACC/AHA Guidelines (https://pubmed.ncbi.nlm.nih.gov/33332149/) and 2021 AHA update (https://pubmed.ncbi.nlm.nih.gov/33853363/) antibiotic prophylaxis is recommended just in patients with high risk for severe infective endocarditis. A good brief discussion about this topic was provided by the following Italian 2020 survey: https://pubmed.ncbi.nlm.nih.gov/32867465/.

Please, add these topics in your Discussion and References, please explain why you chose to give antibiotic prophylaxis to all of your patients. 

Line 153: "two classes (age < 25 and age >=25)" Please, correct with the proper symbol.

Line 195: "26.66 percent (40 patients) were smokers" Please, correct with the symbol %.

Line 224: "Linear regressions showed the strength of association between smoking and oral contraceptives with complications. the strength of association between risk factors and complications has been calculated" Please correct the capital letter.

Author Response

REVIEWER 4

Dear reviewer,

Thank you for your contribution.

Below are the answers to your suggestions.

"Patients with systemic diseases and/or drug therapies known to interfere with soft and hard tissue healing following surgery, have been excluded from the study. The exclusion criteria were pregnancy, have not undergone oral and/or periodontal surgery during the previous 12 months, patients with chronic periodontal disease. Were included non-smokers and patients with healthy dental status. [...] Patients were given antibiotic prophylaxis with 875 mg amoxicillin + 125 mg clavulanic acid two tablets 1 hour before surgery."

According to 2020 ACC/AHA Guidelines (https://pubmed.ncbi.nlm.nih.gov/33332149/) and 2021 AHA update (https://pubmed.ncbi.nlm.nih.gov/33853363/) antibiotic prophylaxis is recommended just in patients with high risk for severe infective endocarditis. A good brief discussion about this topic was provided by the following Italian 2020 survey: https://pubmed.ncbi.nlm.nih.gov/32867465/.

Please, add these topics in your Discussion and References, please explain why you chose to give antibiotic prophylaxis to all of your patients.

It was fixed

Line 153: "two classes (age < 25 and age >=25)" Please, correct with the proper symbol.

It was fixed

Line 195: "26.66 percent (40 patients) were smokers" Please, correct with the symbol %.

It was fixed

Line 224: "Linear regressions showed the strength of association between smoking and oral contraceptives with complications. the strength of association between risk factors and complications has been calculated" Please correct the capital letter.

It was fixed

Reviewer 5 Report

The article "Influence of surgical technique on post-operative complications in the extraction of the lower third molar: A retrospective study" deals with impacted lower third molars. This should be clearly stated in the title, in the second sentence of the abstract, throughout the text, when needed, and in the first sentence of the conclusions. The key words should be modified, as well.

The third sentence of the introduction should be moved right after the first one.

Please do not give abbreviations unless they are used further in the text. Do not give abbreviations explanation twice (see footer of tables 1, 2).

Lines 56-57 are in apparent contradiction with lines 58-60. Please correct. 

Lines 69-70 are a repetition of lines 59-60. Please delete. 

Line 76-77. Please be more specific when defining the aim of the study, mention impacted lower third molar and name the two surgical techniques used. Same for line 234-235.

Line 85-89. The inclusion and exclusion criteria are not properly defined. Please correct. 

Line 94-95. Please give full term for CBCT and IA. 

Line 301 "In the literature we didn’t find uniqueness with respect to age as a risk factor." is in contradiction with line 53-56 "The incidence of postoperative complications and the risks of permanent sequelae  increase with age. Therefore, once the decision to extract an impacted mandibular third  molar has been made, the surgery should be performed as soon as possible and well be fore the age of 24 [5]." Please correct.

Please only give conclusions directly related to your study, not general ones.

The manuscript has to be further checked for correct English. 

Author Response

REVIEWER 5

Dear reviewer,

Thank you for your contribution.

Below are the answers to your suggestions.

The article "Influence of surgical technique on post-operative complications in the extraction of the lower third molar: A retrospective study" deals with impacted lower third molars. This should be clearly stated in the title, in the second sentence of the abstract, throughout the text, when needed, and in the first sentence of the conclusions. The key words should be modified, as well.

Has been added “impacted” where needed in text.

The third sentence of the introduction should be moved right after the first one.

It was fixed.

Please do not give abbreviations unless they are used further in the text. Do not give abbreviations explanation twice (see footer of tables 1, 2).

It was fixed.

Lines 56-57 are in apparent contradiction with lines 58-60. Please correct.

It was fixed.

Lines 69-70 are a repetition of lines 59-60. Please delete.

It was fixed.

Line 76-77. Please be more specific when defining the aim of the study, mention impacted lower third molar and name the two surgical techniques used. Same for line 233-234.

It was fixed.

Line 85-89. The inclusion and exclusion criteria are not properly defined. Please correct.

It was fixed

Line 94-95. Please give full term for CBCT and IA.

It was fixed.

Line 301 "In the literature we didn’t find uniqueness with respect to age as a risk factor." is in contradiction with line 53-56 "The incidence of postoperative complications and the risks of permanent sequelae  increase with age. Therefore, once the decision to extract an impacted mandibular third  molar has been made, the surgery should be performed as soon as possible and well be fore the age of 24 [5]." Please correct.

It was fixed.

Please only give conclusions directly related to your study, not general ones.

The conclusions have been rewritten.

Round 2

Reviewer 1 Report

Dear colleagues!

Thanks for the work you've done. In reality, much has become clear.

About antibiotics, it’s not entirely clear how exactly you made your choice. Especially if we take into account the comorbidity of patients and their physiological characteristics.

I also have a question about parasthesia - how did ST help you with this?

And, most importantly, why did you use such a huge dose of local anesthetic? 6 ml of mepivacaine HCl 2% with 1:100,000 epinephrine used as the local anesthetic agent for inferior alveolar and lingual nerve block (1.8 ml), In association with a buccal injection (1.8 ml).

6 ml, then another 1 carpule and one more. Are you aware of the maximum permissible dose and systemic toxic reaction to the anesthetic when the dosage is increased above the permissible level? Why didn't you determine your patients' body mass index?

Author Response

Kind regards,

Dott. Nicolò Tomizioli

Reviewer 4 Report

The paper can be accepted in the present form.

Author Response

The paper can be accepted in the present form.

Dear reviewer,
Thank you for your contribution.
Kind regards,
Dott. Nicolò Tomizioli

Round 3

Reviewer 1 Report

Hello colleagues!

I am satisfied with your answers.